# Changes in the Physical Function and Psychological Distress from Pre-Implant to 1, 6, and 12 Months Post-Implant in Patients Undergoing Implantable Cardioverter Defibrillator Therapy

**DOI:** 10.3390/jcm9020307

**Published:** 2020-01-22

**Authors:** JinShil Kim, Jin-Kyu Park, Jiin Choi, Sun Hwa Kim, Young Keun On, Mi-Seung Shin, NaYeon Choi, Seongkum Heo

**Affiliations:** 1College of Nursing (Medical campus), Gachon University, Incheon 21936, Korea; kimj317503@gmail.com; 2Division of Cardiology, Department of Internal Medicine, College of Medicine, Hanyang University, Seoul 04763, Korea; 3Office of Hospital Information, Seoul National University Hospital, Seoul 03080, Korea; chocojiin@hanmail.net; 4Department of Nursing, Hanyang University Medical Center, Seoul 04763, Korea; 79ssunhwa@hyumc.com; 5Division of Cardiology, Department of Internal Medicine, Heart Vascular Stroke Institute, Samsung Medical Center, Sungkyunkwan University School of Medicine, Seoul 06351, Korea; yk.on@samsung.com; 6Division of Cardiology, Department of Internal Medicine, Gil Medical Center, Gachon University, College of Medicine, Incheon 21936, Korea; msshin@gilhospital.com; 7Biostatistical Consulting and Research Lab, Medical Research Collaborating Center, Hanyang University, Seoul 04763, Korea; nayeon@hanyang.ac.kr; 8Georgia Baptist College of Nursing, Mercer University, Atlanta, GA 31207, USA; Heo_s@mercer.edu

**Keywords:** physical function, anxiety, depression, implantable cardioverter-defibrillator

## Abstract

Recipients of implantable cardioverter-defibrillator (ICD) therapy in Western countries often experience distressful physical and psychological adjustments. Sociocultural influences on post-implant recovery are likely; however, evidence from other ethnic/cultural backgrounds is lacking. This study aimed to examine the changes in physical function and psychological distress (anxiety and depressive symptoms) from pre-implant to one, six, and 12 months post-implant in Korean patients undergoing ICD therapy. A total of 34 patients underwent pre- to post-implant longitudinal assessments of physical and psychological function using mixed modeling procedures. Physical function significantly declined from pre-implant to one month post-implant (B = −10.05, *p* = 0.004) and then nearly returned to the pre-implant level at six months post-implant (B = 8.34, *p* = 0.028). This level of improvement continued through 12 months post-implant. In psychological distress, significant improvements were observed from pre-implant to one month (anxiety (B = −1.20, *p* = 0.020) and in depressive symptoms (B = −1.15, *p* = 0.037)), which then plateaued without significant changes from one to 12 months. We concluded that physical function recovery occurred six months post-implant, but function remained poor until 12 months post-implant. Psychological distress improved one month post-implant and it was maintained. Clinicians must provide more intensive interventions to improve long-term physical function after ICD therapy.

## 1. Introduction

Implantable cardioverter-defibrillators (ICDs) for primary or secondary prevention of sudden cardiac death have proven long-term survival benefits over conventional therapeutic regimens [1,2]. Furthermore, their utilization in older adults (aged 70 years or older) has increased more than ever [3]. ICD therapy might lead to physical and psychological maladjustment issues, such as physical functional limitation and psychological distress of anxiety and depressive symptoms [4,5,6,7]. Post-implant psychological recovery seems to not occur naturally over time, while psychological distress, such as anxiety and depressive symptoms, seems to not dissipate after the ICD procedure. More than half of the patients who were anxious at the time of implantation have chronic anxiety 12 months later [8]. Anxious ICD recipients had a fourfold higher risk of death than those without anxiety [9]. A minority of ICD recipients (14%) also presented persistent depressive symptoms three months post-implantation [10].

Despite the extensive use of ICD therapy in Western countries, it remains underutilized in Korea, particularly for primary prevention [11]. Post-implant physical and psychological recovery has also rarely been addressed. Post-implant recovery guidelines, which may vary by an individuals’ cardiac condition and testing results of ventricular arrhythmias at follow-up, instruct patients to expect four to eight weeks to return to normal life and resume physical activity [12,13]. In addition to restrictions on physical activities for a certain period following ICD implantation, underlying cardiac conditions, particularly heart failure that is a predominant condition in ICD recipients for primary prevention, might further challenge the post-implant physical function recovery. Sociocultural influences on the post-implant trajectory (adjusting to the ICD and returning to normal life) are likely to be present [14]. However, evidence of these influences in other sociocultural environments is limited. In particular, pre- to post-implant physical and psychological function recovery is poorly understood in Korea. Furthermore, the illness trajectories of primary and secondary ICD recipients may affect both physical and psychological function, and each patient will experience a different post-implant physical and psychological function recovery pattern. Evidence-based data that are related to physical and psychological recovery are also limited. Thus, findings regarding changes in physical and psychological function of Korean patients undergoing ICD therapy may provide insight into the management of the physical and psychological status of patients receiving ICD therapy in East Asian environments. The purposes of this study were to (1) examine the prevalence of physical functional limitations and psychological distress; (2) examine pre- to 1, 6, and 12 months post-implant changes in the physical and psychological function of newly implanted patients; and, (3) compare those changes between patients receiving primary and secondary ICD therapy.

## 2. Methods

### 2.1. Study Design and Procedure

Using a longitudinal study deign, repeated measurements on pre- and 1, 6, and 12 months post-implant physical and psychological distress of anxiety and depressive symptoms were conducted from newly implanted patients who underwent ICD therapy for primary or secondary prevention of sudden cardiac death. Approvals for this study protocol were obtained from the institutional review boards of the university-affiliated hospitals (Ethical approval codes: HYUH 2016-08-040-003; SMC2017-04-111). A written informed consent process was followed prior to data collection. Adhering to the study protocol that indicated detailed instructions about the study, a graduate nursing student and a clinical research coordinator collected the data through face-to-face interviews when admitting patients to the hospitals for ICD implantation (pre-implant) and at 1, 6, and 12 months post-implant outpatient visits for regular follow-ups.

### 2.2. Subjects

Patients participated in this study if they met the following criteria: (1) age 21 years or older and (2) satisfying the ICD implantation criteria for primary or secondary prevention of sudden cardiac death [15,16]. Patients were not able to participate if they met one of the following criteria: (1) presence of a bi-ventricular pacemaker or (2) documented neurocognitive disorder, such as dementia or Alzheimer’s disease or a mental disorder.

### 2.3. Measures

Physical function: The Korean Activity Status Index (KASI), the Korean version of the Duke Activity Status Index [17], is a measure of physical function [18]. Fifteen daily physical activities were selected while considering environmental and cultural contexts, with each activity having a weighted value assigned by the energy expenditure required for each [18]. The possible scores range from 0 to 79.0, with higher scores indicating better physical function and physical functional classes that range from I (≥46) to IV (˂4). A physical functional limitation was identified when the score was less than 46. The reliability and validity of the KASI were documented previously [18].

Anxiety and depressive symptoms: The Hospital Anxiety and Depression Scale (HADS) is a measure of psychological distress of anxiety and depressive symptoms [19]. The HADS consists of 14 items with four-point Likert response options (0 to 3), which are evenly divided into anxiety and depression subscales. The possible scores range from 0 to 21 for each subscale, with higher scores indicating more severe anxiety or depressive symptoms. Scores of 8 or higher are often used to define psychological distress [20,21].

Demographic and clinical data were also obtained. For clinical information, reviews of electronic medical records were performed, and data, including ICD indication, date of ICD implantation, left ventricular ejection fraction, medical history, and prescribed medication, were extracted. The functional severity that was imposed by the cardiac conditions was assessed using the New York Heart Association (NYHA) classification of I (asymptomatic) to IV (severely symptomatic).

### 2.4. Statistical Analysis

Descriptive statistics, including mean ± SD for normally distributed numerical variables, median (Q1–Q3) for non-normally distributed numerical variables, and frequency (n) and percentage (%) for categorical variables, were computed to describe the sample characteristics. Group comparisons (primary and secondary ICD recipients) were performed using the independent t-test, Wilcoxon rank-sum test, and chi-squared test or Fisher’s exact test, depending on the levels or distributions of the variables. Frequency and percentage were computed based on the predetermined cut-off points for the prevalence of functional limitation and psychological distress. Wilcoxon rank-sum tests were also conducted at each time point to compare the levels of physical function and anxiety and depressive symptoms between primary and secondary ICD recipients. Lastly, mixed modeling procedures were performed to examine the group and time effects for the pre-implant to 1-, 6-, and 12-month post-implant changes in physical and psychological function. The mixed modeling procedures incorporated different sample sizes at each time point during statistical analyses typically performed for this type of a repeated-measures statistical analysis. The inclusion of incomplete data is more likely to reflect ICD recipients’ post-implant recovery over time. The statistical analyses were performed while using SAS version 9.4 [22]. The level of significance was set at < 0.05.

## 3. Results

### 3.1. Baseline Characteristics

A total of 34, 29, 32, and 31 patients completed the pre-to post-implant assessments of physical function and psychological distress at baseline (pre-implant) and at 1, 6, and 12 months after ICD implantation. The reasons for attrition during the follow-up period were missed appointments, lack of interest, or time constraints. Table 1 presents the baseline sample characteristics. The mean age was 56.2 (±12.0) years at baseline. The majority of patients who completed the baseline interviews were male (73.5%) and married (85.3%). Half of the ICD recipients had asymptomatic functional severity with NYHA class I (51.5%).

Nineteen (55.9%) and 15 (44.1%) patients received ICDs for primary and secondary prevention of sudden cardiac death, respectively (Table 1). The primary ICD recipients were older (60.3 vs. 51.0 years; *p* = 0.023), had a lower median LVEF (29.0% vs. 58.0%; *p* = 0.001), had a higher prevalence of myocardial infarction (47.4% vs. 6.7%; *p* = 0.020), and were more likely to be taking diuretics (79.0% vs. 40.0%; *p* = 0.020) and statins (68.4% vs. 20.0%; *p* = 0.005) than secondary ICD recipients.

### 3.2. Physical Function and Psychological Distress of Primary Versus Secondary ICD Therapy at Pre-Implant and 1, 6, and 12 months Post-Implant

Table 2 compares the physical function and psychological distress of primary and secondary ICD recipients by time point. For the entire sample, median physical functional scores over time ranged from 35.2 at one-month post-implant to 50.6 at pre-implant. The median anxiety scores ranged from 2.0 at 1-month post-implant to 5.0 at pre-implant. The median depressive symptom scores ranged from 2.0 at 12-month post-implant to 4.0 at pre-implant. The only significant intergroup difference was found in pre-implant physical function, with secondary ICD recipients presenting better physical function (38.8 vs. 57.8; *p* = 0.026) than the primary ICD recipients.

Using KASI scores < 46 for physical functional limitation and HADS scores ≥ 8 for clinically significant anxiety and depressive symptoms as cut-off points, the prevalence of pre-implant to 1-, 6-, and 12-month post-implant physical and psychological distress was 41.2%, 69.0%, 40.6%, and 48.4% for physical function; 17.6%, 20.7%, 12.5%, and 13.3% for anxiety; and, 20.6%, 10.3%, 12.5%, and 13.3% for depressive symptoms, respectively (Figure 1). There were no associations between physical or psychological distress and ICD indications of primary and secondary prevention, except for baseline physical function, with primary ICD recipients having worse physical function than secondary ICD recipients (*x*^2^ = 4.97, *p* = 0.026).

### 3.3. Changes in Physical Function and Psychological Distress from Pre-Implant to 12 months Post-Implant

A series of mixed modeling procedures was performed to examine the changes in physical function, anxiety, and depressive symptoms over time (Table 3), and further compare these changes between primary and secondary ICD recipients (Table 4). After controlling for LVEF in the physical function model due to its influence on physical function, but not on anxiety and depressive symptoms, a significant physical functional decline was noted from pre-implant to one-month post-implant (B = −10.05; *p* = 0.004). The value significantly improved at six months post-implant as compared to that at one month post-implant (B = 8.34; *p* = 0.028) and then plateaued until 12 months post-implant (B = 0.18, *p* = 0.932) (Table 3). Anxiety and depressive symptoms were both significantly improved from pre-implant to one month post-implant (anxiety: B = −1.20, *p* = 0.020; depressive symptoms: B = −1.15, *p* = 0.037) and then the improved anxiety and depressive symptoms continued until six months and 12 months post-implant (anxiety: B = 0.80, *p* = 0.324 at one to six months and B = −1.03, *p* = 0.050 at six to 12 months post-implant; depressive symptoms: B = −0.03, *p* = 0.969 at one to six months and B = −0.20, *p* = 0.661 at six to 12 months post-implant). In the models including primary and secondary ICD, no significant intergroup differences were noted in physical and psychological function over time (Table 4).

## 4. Discussion

This study first examined pre-implant to 12-month post-implant physical and psychological recovery of patients undergoing primary and secondary ICD therapy recruited from a metropolitan area in South Korea. The patterns of physical and psychological recovery from pre-implant to 12 months post-implant differed considerably, while there was no group effect on the post-implant physical and psychological recovery between patients with ICD for primary or secondary prevention of sudden cardiac death. Physical function worsened one month post-implant when compared to pre-implant, but improved six months post-implant. The improvement continued until 12 months post-implant. Anxiety and depressive symptoms improved one month post-implant compared to those at pre-implant, and the improvements continued until 6 and 12 months post-implant. Thus, post-implant recovery seemed to occur in 6 months for physical function and one month for psychological distress.

ICD implantation commonly involves an ongoing transitional process, in which ICD recipients may undergo early distressful changes in their daily life and then adjust to and modify their lifestyles accordingly [5]. While post-implant recovery and adjustment to normal life are likely to be under sociocultural and environmental influences [14], post-implant guidelines were developed based on evidence from Western countries [12,23]. Thus, the exploration and understanding of the physical and psychosocial experiences of patients in regions other than Western countries, such as Asian countries, are required for providing care that considers their actual needs following ICD implantation rather than care that is based on general management strategies. In this study, pre- and post-implant physical functional adjustment of newly implanted patients in Korea was initially examined over 12 months. Overall, the physical function of ICD recipients was substantially poor, regardless of ICD indication (KASI scores: 35–50 out of 79), with the prevalence of physical functional limitation being the highest at one month post-implant (69.0%). Physical function returned to the pre-implant level at six months post-implant (40.6%; baseline: 41.2%) and plateaued until 12 months post-implant (48.4%). Although the prevalence of physical functional limitation could not be compared among studies due to differences in population characteristics, measurement tools, or definitions, substantial physical impairment was noted in the U.S., with 35–43% of older patients with ICDs (mean age, 71.1 years) being followed for 18 months [24]. Thus, physical function needs to be assessed and managed to reduce early physical functional limitations after ICD implantation to facilitate faster recovery of physical function.

In this study, changes in physical function from pre-implant to one month post-implant and from one month to six months post-implant remained significant after controlling for LVEF. Poor physical function was previously reported, while the trajectory seemed to show subtle differences. In one study, patients (mean age, 60.6 years) demonstrated poor physical function over time with 35.98 at pre-implant, 36.72 at 6 weeks post-implant, 37.10 at 6 months post-implant, and 37.64 of 100 at 12 months post-implant on the Physical Functioning subscale of the Medical Outcomes Survey Short Form-36 (SF-36), which seemed to show that there were no noticeable changes [25]. Furthermore, ICD recipients who had ICDs for 1.35 years (mean baseline age, 64.9 years), including 47% primary and 53% secondary ICD recipients, had poor physical functional scores on the SF-36 subscale, which persisted for 12 months (52.37 of 100 at baseline and 56.12 of 100 at 12 months post-implant) [26]. The findings of this study and prior studies imply that poor physical function seemed to be substantial, regardless of ICD indication, and persisted after the post-implant surgical recovery period [25,26]. Therefore, ongoing assessment and management of physical function are needed before and after ICD implantation.

The presumable influences of ICD indication on physical function were not significant at most of the time points in this study, with a significant intergroup difference found prior to implantation only. Both of the groups showed similar decline at 1 month post-implant (primary vs. secondary: ICD 35.8 vs. 34.0; *p* = 0.842); then, physical function at 6 months post-implant improved when compared to that at 1 month post-implant (primary vs. secondary: 46.8 vs. 46.8; *p* = 0.366). Conversely, in one study that compared physical and psychological scores by ICD indication over 6 months in patients who underwent ICD implantation, patients receiving primary ICD therapy (53%) (mean age = 66.3 years) had significantly lower physical scores on the Physical Functioning Subscale of the SF-36 than patients receiving secondary ICD therapy (47%) (mean age = 64.5 years) (62.7 vs. 71.9; *p* = 0.002) [9]. The small subgroup samples in this study might have influenced the non-significant differences in physical function between the two groups. The younger age of our patients might also explain this inconsistent result, in that physical recovery from ICD procedure-related restrictions in physical activities occurred 6 months post-implant in both primary and secondary ICD recipients in this study, while implantation improved the physical function of primary ICD recipients to a better state than pre-implant. Further studies are needed to examine the physical status of both groups. In addition, the findings of this study and prior studies imply that the assessment and management of physical function are required before and after implantation, regardless of the purpose of the procedure.

According to the general instructions for physical activity after ICD implantation, patients can perform daily activities in 1 month and regain their prior functional state in a couple of months [12,13]. Post-implant recovery of physical function in Korean patients seems to take a little longer. The pre-implant physical functional level was restored within 6 months; however, physical function at this time was still poor. This poor physical function plateaued at 12 months, with approximately half of the patients having limited physical function over time that could be imposed by the underlying cardiac disease itself. The findings of this study and prior studies show that the ICD implantation procedure is unlikely to instigate additional functional restrictions after the initial 4-week to 6-month post-implant transitional process. Thus, physical functional support should be continued as part of the underlying disease management beyond the physical activity and exercise guidelines that are associated with the ICD procedure. In addition, decreased physical function has predictive value for clinical outcomes, including survival [27]. Thus, constant surveillance and efforts to improve physical function in routine care are required.

Psychological distress is a major concern following ICD implantation. Anxiety and depressive symptoms are both prevalent in Western countries [5,28,29]; however, post-implant psychological symptoms in Korea have not been well addressed. In this initial longitudinal assessment of anxiety and depressive symptoms, the pre-implant prevalence of anxiety and depressive symptoms was 17.6% and 20.6%, respectively. The prevalence of anxiety in this study was considerably lower than that in a prior ICD study: 67.5% pre-implant, 49.9% 6 weeks post-implant, 47.8% 6 months post-implant, and 56.7% 12 months post-implant [25]. For chronic anxiety, the prevalence was 53.9% 12 months post-implant [8]. In the Hallas et al. study and this study, the mean subject age (60.6 vs. 56.2 years), male sex (86.5% vs. 73.5%), and married status (73.1% vs. 85.3%) were similar. The prevalence of depressive symptoms in this study was similar to that in prior ICD studies: prevalence at pre-implant and 6 weeks, 6 months, and 12 months post-implant was 23.0%, 26.9%, 22.9%, and 9.0%, respectively (no statistical comparisons) [25], and 14% had persistent depression at three months post-implant [10], which was defined as a HADS depression subscale score of 8 or higher. Thus, further studies are needed to determine the reasons for the differences in the prevalence of anxiety while considering the possible cultural effects on psychological status.

In this study, anxiety and depressive symptoms significantly improved from pre-implant to 1 month post-implant, and the improvements continued through 6 and 12 months post-implant. These patterns are different from those in prior studies. In an ICD study (mean age, 60.6 years; 86.5% male; and, 73.1% married status), changes in the mean scores of depressive symptoms assessed by the HADS from pre-implant to six weeks, six months, and 12 months post-implant were not statistically significant [25]. The findings of another ICD study (mean age, 55.2 years; 73.5% male; and, 63.2% married status) were consistent with the findings of the Hallas et al. study, showing no significant changes in anxiety assessed by the State-Trait Anxiety Inventory and depressive symptoms assessed by the Centre for Epidemiologic Studies Depression Scale over pre-implant and 1, 6, and 12 months post-implant [30]. The mean age and sex ratio in this study and prior studies were very similar. In this study and the Hallas et al. study [25], the ratios of married status were also very similar, while those in the Kamphuis et al. study [30] were a little bit lower. The instruments that were used in this study and the Hallas et al. study to assess depressive symptoms were the same, but the change patterns differed. In contrast, the Hallas et al. [25] and Kamphuis et al. [30] studies used different instruments to assess depressive symptoms, but the change patterns were similar. The region of interest is one difference between this study and the prior studies. Our study was conducted in an Asian country, while the two prior studies were conducted in Western countries and the differences in findings imply that culture might influence changes in psychological status after ICD implantation. Thus, further studies are needed to examine changes in depressive symptoms and anxiety while considering the instruments, sample characteristics, and regions.

Our study also compared psychological distress between patients receiving ICD therapy for primary and secondary prevention of sudden cardiac death, which has not been frequently examined. Hypothetically, patients receiving ICD therapy for primary prevention might have greater difficulty in adapting physically to the ICDs because of their sicker state and functional limitations, while patients receiving ICD therapy for secondary prevention might also have poor psychological adaptation because of traumatic survival from the previous experience of a potentially fatal dysrhythmia. However, differences in physical and psychological status and long-term changes after ICD implantation between the two groups have rarely been examined. In this study, although there were no significant intergroup differences in physical and psychological distress or in change patterns after ICD implantation, both of the groups had different clinical contexts, prior experiences, and reasons for ICD implantation. Thus, further studies are needed to validate the findings of this study in larger samples considering possible factors affecting physical and psychological status.

## 5. Limitations

The small sample size was one of the major limitations of this study. Because of this limitation, the study results need validation in different regions with larger samples. Even though the sample size is small, while considering that little is known about the physical and psychological changes over time in both patients with primary and secondary ICD, especially in Asian countries, we believe that the findings can still contribute to the body of existing knowledge as preliminary data. Furthermore, factors that possibly affect post-implant physical and psychological recovery were not addressed, because it was beyond the scope of this study. Thus, examining the prevalence and changes in physical and psychological status before and after ICD implantation over time in larger samples and while controlling for possible covariates may be beneficial.

## 6. Conclusions

Patients experienced more physical functional limitations (40.6–69.0%) than psychological distress (10.3–20.7%) over time in this initial examination of pre- and post-implant changes in physical function and psychological distress occurring in patients with ICDs in Korea. The change patterns in physical function and psychological distress from pre-implant to 12 months post-implant also differed. Post-implant physical function worsened at 1 month post-implant, but recovered 6 months post-implant and continued until 12 months post-implant. Post-implant psychological distress decreased at 1 month post-implant and these improvements continued until 12 months post-implant. Interestingly, primary and secondary ICD recipients showed similar change patterns in the physical function and psychological distress over time.

## Figures and Tables

**Figure 1 jcm-09-00307-f001:**
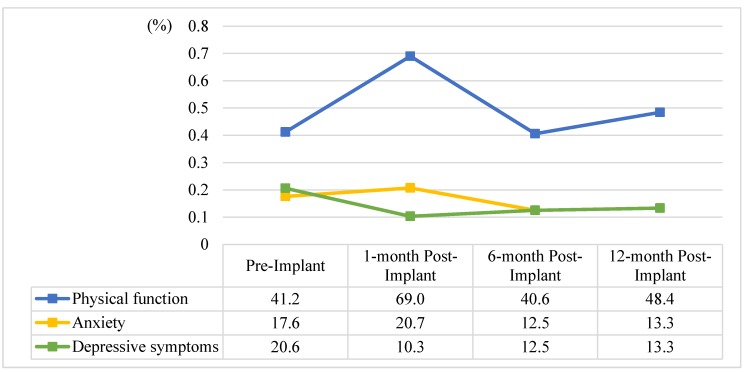
The prevalence of physical functional limitation, anxiety, and depressive symptoms in patients with implantable cardioverter-defibrillators. ICD, implantable cardioverter-defibrillator. Note: Use the Korean Activity Status Index scores < 46 for physical functional limitation and the Hospital Anxiety and Depression Scale scores ≥ 8 for clinically significant anxiety and depressive symptoms as cut-points.

**Table 1 jcm-09-00307-t001:** Baseline Demographic and Disease-related Characteristics of Patient Groups.

Variables	Total (*n* = 34)	Primary ICD (*n* = 19)	Secondary ICD (*n* = 15)	*p*-Value
	Mean ± SD/Median (Q1–Q3) or N (%)			
Age (years)	56.2 ± 12.0	60.3 ± 10.3	51.0 ± 12.4	0.023
Sex (male)	25 (73.5)	13 (68.4)	12 (80.0)	0.697
Marital status				
Married	29 (85.3)	18 (94.7)	11 (73.3)	0.210
Never married	2 (5.9)	0 (0.0)	2 (13.3)	
Divorced	3 (8.8)	1 (5.3)	2 (13.3)	
Education				
<High school	11 (32.4)	8 (42.1)	3 (20.0)	0.429
=High school	6 (17.7)	3 (15.8)	3 (20.0)	
≥College	17 (50.0)	8 (42.1)	9 (60.0)	
LVEF (%)	33.5 (28.0–58.0)	29.0 (26.0–33.0)	58.0 (42.9–63.0)	0.001
NYHA class (*n* = 33)				
I	17 (51.5)	11 (57.9)	6 (42.9)	0.781
II	13 (39.4)	7 (36.8)	6 (42.9)	
III	2 (6.1)	1 (5.3)	1 (7.1)	
IV	1 (3.0)	0 (0.0)	1 (7.1)	
Comorbidities (yes)				
Myocardial infarction	10 (29.4)	9 (47.4)	1 (6.7)	0.020
Hypertension	13 (38.2)	10 (52.6)	3 (20.0)	0.052
Atrial fibrillation	7 (20.6)	5 (29.4)	2 (13.3)	0.403
Diabetes mellitus	6 (17.6)	3 (15.8)	3 (20.0)	>0.999
Prescribed medicine				
Amiodarone	8 (23.5)	6 (31.6)	2 (13.3)	0.213
Aspirin	8 (23.5)	6 (31.6)	2 (13.3)	0.213
ACEi	8 (23.5)	5 (26.3)	3 (20.0)	0.666
ARB	15 (44.1)	11 (57.9)	4 (26.7)	0.069
BB	21 (61.8)	14 (73.7)	7 (46.7)	0.107
Diuretics	21 (61.8)	15 (79.0)	6 (40.0)	0.020
Statins	16 (47.0)	13 (68.4)	3 (20.0)	0.005

ICD, implantable cardioverter-defibrillator; LVEF, left ventricular ejection fraction; NYHA, New York Heart Association; ACEi, angiotensin-converting enzyme inhibitor; ARB, angiotensin II receptor blocker; BB, beta-blocker. Normally distributed numerical variables are shown as mean ± SD and were tested by independent t-test; non-normally distributed numerical variables are shown as median (Q1–Q3) and were tested by the Wilcoxon rank-sum test; and categorical variables are shown as *n* (%) and were tested by the Chi-squared test or Fisher’s exact test.

**Table 2 jcm-09-00307-t002:** Pre-Implant and 1, 6, and 12 months Post-Implant Physical and Psychological Dysfunction Scores in Primary versus Secondary ICD Recipients.

Variables	Time Points	*n*	Total	Primary ICD	Secondary ICD	*p*-Value
Physical function	Pre	34	50.6 (37.8–67.8)	38.8 (32.3–59.8)	57.8 (46.3–76.8)	0.026
	1 month	29	35.2 (27.3–57.3)	35.8 (27.3–51.8)	34.0 (26.3–65.05)	0.842
	6 months	32	46.8 (38.0–62.6)	46.8 (29.3–59.8)	46.8 (38.8–67.8)	0.366
	12 months	31	46.3 (33.3–62.3)	43.8 (32.3–59.8)	58.3 (41.3–67.8)	0.109
Anxiety	Pre	34	5.0 (2.0–7.0)	5 (2–7)	5 (1–7)	0.688
	1 month	29	2.0 (1.0–7.0)	2 (1–4)	3 (1–8)	0.324
	6 months	30	4.0 (2.0–6.0)	5 (2–7)	4 (2–6)	0.969
	12 months	30	2.5 (1.0–6.0)	4 (1–6)	2 (0–7)	0.572
Depressive symptoms	Pre	34	4.0 (3.0–7.0)	4 (3–7)	4 (2–8)	>0.999
	1 month	29	3.0 (1.0–6.0)	3 (1–5)	3 (0–7)	>0.999
	6 months	30	3.0 (1.0–5.0)	3 (1–6)	1 (1–4)	0.382
	12 months	30	2.0 (1.0–6.0)	4 (1–6)	1 (0–8)	0.446

Non-normally distributed numerical variables are presented as median (Q1-Q3) and were tested by the Wilcoxon rank-sum test.

**Table 3 jcm-09-00307-t003:** Mixed Modeling Procedures: Time Effects on Pre-Implant-to 1, 6, and 12 months Post-Implant Changes in Physical Function and Psychological Distress of Defibrillator Recipients.

Variables	Effect		B	SE	95% CI	*p*-Value
Physical function	Intercept		31.90	7.28	17.13, 46.66	<0.0001
	LVEF		0.41	0.17	0.07, 0.74	0.019
	Time	0 vs. 1 mo	−10.05	3.20	−16.57, −3.52	0.004
		1 vs. 6 mo	8.34	3.62	0.96, 15.72	0.028
		6 vs. 12 mo	0.18	2.15	−4.19, 4.56	0.932
Anxiety	Intercept		4.68	0.52	3.61, 5.74	<0.0001
	Time	0 vs. 1 mo	−1.20	0.49	−2.19, −0.20	0.020
		1 vs. 6 mo	0.80	0.80	−0.83, 2.43	0.324
		6 vs. 12 mo	−1.03	0.50	−2.06, 0.00	0.050
Depressive symptoms	Intercept		4.76	0.53	3.70, 5.83	<0.0001
	Time	0 vs. 1 mo	−1.15	0.53	−2.22, −0.08	0.037
		1 vs. 6 mo	−0.03	0.79	−1.64, 1.58	0.969
		6 vs. 12 mo	−0.20	0.45	−1.12, 0.72	0.661

SE, standard error; CI, confidence interval; mo, month; LVEF, left ventricular ejection fraction. An effect of LVEF was examined in a model of physical function only based on its significant association with physical function but not with anxiety and depressive symptoms.

**Table 4 jcm-09-00307-t004:** Mixed Modeling Procedures: Group-by-Time Effects on Pre-Implant-to 1, 6, and 12 months Post-Implant Changes in Physical Function and Psychological Distress of Defibrillator Recipients.

Variables	Effect		B	SE	95% CI	*p*-Value
Physical function	Intercept		31.15	11.01	8.82, 53.48	0.008
	LVEF		0.42	0.20	0.01, 0.83	0.045
	Group (primary)		0.58	6.33	−12.27, 13.43	0.928
	Time	0 vs. 1 mo	−10.05	3.20	−16.57, −3.52	0.004
		1 vs. 6 mo	8.34	3.62	0.96, 15.72	0.028
		6 vs. 12 mo	0.19	2.15	−4.19, 4.56	0.932
Anxiety	Intercept		4.67	0.75	3.16, 6.18	<0.0001
	Group (primary)		0.01	0.96	−1.94, 1.96	0.992
	Time	0 vs. 1 mo	−1.20	0.49	−2.19, −0.20	0.020
		1 vs. 6 mo	0.80	0.80	−0.83, 2.43	0.324
		6 vs. 12 mo	−1.03	0.50	−2.06, 0.00	0.050
Depressive symptoms	Intercept		4.74	0.76	3.21, 6.27	<0.0001
	Group (primary)		0.04	0.98	−1.94, 2.03	0.965
	Time	0 vs. 1 mo	−1.15	0.53	−2.22, −0.08	0.037
		1 vs. 6 mo	−0.03	0.79	−1.64, 1.58	0.969
		6 vs. 12 mo	−0.20	0.45	−1.12, 0.72	0.661

Recipients. SE, standard error; CI, confidence interval; mo, month; LVEF, left ventricular ejection fraction.

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
