# Peer review of "Changes in the Physical Function and Psychological Distress from Pre-Implant to 1, 6, and 12 Months Post-Implant in Patients Undergoing Implantable Cardioverter Defibrillator Therapy"

_jcm, 2020, doi:10.3390/jcm9020307_

Round 1

Reviewer 1 Report

Although data are well presented, I think that a further increase of sample size could make more interesting this manuscript.

Author Response

We fully agree with the reviewer’s concern. Given a research design that conducted repeated measurements taken from individuals, increasing a sample size is challenging. Instead, we acknowledged this as a major limitation as follows:

One of the major limitations of this study was its small sample size. Because of this limitation, the study results need validation in different regions with larger samples.

Reviewer 2 Report

Dear authors,

thank your for revising the dcoument and adding the limitation of the small sample size.

Author Response

Thank you for your kindness.

Round 2

Reviewer 1 Report

None

This manuscript is a resubmission of an earlier submission. The following is a list of the peer review reports and author responses from that submission.

Round 1

Reviewer 1 Report

Kim and colleagues present data about physical and pschological parameters before and after ICD implantation.

Major comments:

A total number of 34 patients is counted. In the Results section only 29 patients answered the 1mo FU questionnaire and 31 the 12 mo FU questionnaire. I would suggest to count only patients who have answered all of the questionnaires. This is supposed to be less than 29. The reasons for not answering the questionnaire should be described.

In the patient’s characteristics 57.9% of patients receiving ICD for primary prevention of SCD were NYHA class I. Could the authors please comment on the indication für ICD implantation for primary prevention of SCD because European guidelines only indicate an ICD for NYHA ³II.

Table 1: I would change „Medical problems (yes)“ to „Comorbidities“

Table 1: Medication is only available for 33 patients. For this small group a complete description of medication for all patients is mandatory.

It is not clear why patients for primary and for secondary preventaion are compared with each other. I would recommend to strictly divide both groups because the time course is interesting not the differences between groups. The secondary prevention group was totally different looking at the ejection fraction.

I would recommend to show the time courses for the questionnaires for primary prevention spearated from secondary prevention.

Why does physical function differs in that way? Better after 1 month when ICD healing has just finished? Is this not contradictory?

Page 2, line 76: seems to be „to be present“ instead of „to present“.

Reviewer 2 Report

The manuscript “Changes in the Physical Function and Psychological Distress from Pre-Implant to 1-, 6-, and 12-Month Post-Implant in Patients with Implantable Cardioverter-Defibrillator Therapy “ have enrolled a group of 34  Korean patients.

The main findings odf this study are:

post-implant physical function recovery occurred at 6-month post-implant, but the function still poor until 12- month post-implant. psychological distress improved at 1-month post-implant and maintained.

The topic of manuscript is interesting but I have doubts in this current form.

Although data are well presented the methodology and low sample size don’t permit to have a robust clinical message. The main suggestion is to increase sample size.